



# Standardized flux seasonality metrics: A companion dataset for FLUXNET annual product

Linqing Yang[1, *], Asko Noormets[1, *]

[1]Department of Ecology and Conservation Biology, Texas A&M University, College Station, TX, USA

*Correspondence: linqingyang_bnu@tamu.edu (technical questions), noormets@tamu.edu (general questions)

**Abstract:** Phenological events are integrative and sensitive indicators of ecosystem processes that respond to climate, water and nutrient availability, disturbance, and environmental change. The seasonality of ecosystem processes, including biogeochemical fluxes, can similarly be decomposed to identify key transition points and phase durations, which can be determined with high accuracy, and are specific to the processes of interest. As the seasonality of different processes differ, it

can be argued that the interannual trends and responses to environmental forcings can be better described through the fluxes' own temporal characteristics than through correlation to traditional phenological events like bud-break or leaf coloration. Here we present a global dataset of seasonality or phenological metrics calculated for gross primary productivity (GPP), ecosystem respiration (RE), latent heat (LE) and sensible heat (H) calculated for the FLUXNET 2015 Dataset of about 200 sites and 1500 site-years of data. The database includes metrics (i) on absolute flux scale for comparisons with flux magnitudes, and (ii) on

normalized scale for comparisons of change rates across different fluxes. Flux seasonality was characterized by fitting a single-pass double-logistic model to daily flux integrals, and the derivatives of the fitted time series were used to extract the phenological metrics marking key turning points, season lengths and rates of change. Seasonal transition points could be determined with 95% confidence interval of 6-11 days for GPP, 8-14 days for RE, 10-15 days for LE and 15-23 days for H. The phenology metrics derived from different partitioning methods diverged, at times significantly.

This Flux Seasonality Metrics Database (FSMD) can be accessed at U.S. Department of Energy's (DOE) Environmental Systems Science Data Infrastructure for a Virtual Ecosystem (ESS-DIVE, https://data.ess-dive.lbl.gov/view/doi:10.15485/1602532; Yang and Noormets, 2020). We hope that it will facilitate new lines of research, including (1) validating and benchmarking ecosystem process models, (2) parameterizing satellite remote sensing phenology and Phenocam products, (3) optimizing phenological models, and (4) generally expanding the toolset for interpreting

ecosystems responses to changing climate.

## 1 Introduction

Phenology, the timing of life-cycle events and phases of plants and animals, and their relationship with the environment, especially climate (Lieth, 1974;Piao et al., 2019), is an important indicator of ecosystem dynamics. It is an integrating record of the effects of global warming and other environmental changes on biological processes (Noormets et al., 2009;Post and





Stenseth, 1999). Current phenology studies focus primarily on structural changes such as bud break, flowering, leaf coloring, and leaf fall. However, the functional aspects of plant activities, although invisible, also provide quantitative measures of plant responses to changes in environmental conditions and underlie the structural changes (Fitzjarrald et al., 2001;Schwartz, 2003;Schwartz and Crawford, 2013).Ecosystem processes such as biogeochemical fluxes also show seasonal changes and can be decomposed to key transition dates and phase durations, that characterize the exchanges of mass and energy between plants

and the environment, and may exert mutual feedback (Baldocchi et al., 2001;Freedman et al., 2001).

Currently, the phenology datasets mainly have three sources: (i) ground-based observations of plant structural changes, (ii) camera-based observations of canopy reflectance (or near-surface remote sensing observations), and (iii) satellite-based observations of land surface reflectance. The ground-based phenology is a traditional but very useful method in phenology studies. For example, the USA National Phenology Network (USA-NPN) has collated observations of first bloom and first

leaf of lilac and honeysuckle from the 1960s across the Contiguous United States (Schwartz et al., 2012;CONUS, United States territory, not including Hawaii or Alaska; Betancourt et al., 2007;Glynn and Owen, 2015). The USA-NPN was established in part to assemble long-term phenology datasets for a broad array of species across the United States, which can be used to determine the extent to which species, populations, and communities are vulnerable to ongoing and projected future changes in climate (Glynn and Owen, 2015;Schwartz et al., 2012). The camera-based phenology observations such as the Phenocam

network (https://phenocam.sr.unh.edu/webcam/; Richardson et al., 2018;Richardson, 2019) use high-resolution digital cameras to characterize canopy phenology through the color information from the images (Brown et al., 2016;Richardson et al., 2018). The remote sensing has been used to detect vegetation green-up and canopy development (Ganguly et al., 2010;Julien and Sobrino, 2009;Zhang et al., 2003;Zhang et al., 2018). While the remote-sensing-based phenology product estimates transition dates from a continuous reflectance time series, it is truthed against ground-based event data. The seasonality of ecosystem

processes, including biogeochemical fluxes, can similarly be decomposed to identify key transition points and phase durations, which can be determined with high accuracy, and are specific to the processes of interest. As the seasonalities of different processes differ, it can be argued that the interannual trends and responses to environmental forcings can be better described through the fluxes' own temporal characteristics than through correlation to traditional phenological events like bud-break or leaf coloration.

Therefore, the objective of this paper is to generate an objective and standardized flux seasonality metrics dataset, which can act as the companion dataset for the FLUXNET product. This study aims to develop a comprehensive framework to studying the seasonality of ecosystem processes systematically with eddy covariance flux measurements including gross primary productivity (GPP), ecosystem respiration (RE), latent heat (LE), and sensible heat (H). As ecosystem and Earth System Models are increasingly tested and developed using the very high temporal and increasing spatial coverage of eddy covariance

sites, the added information of explicit and standardized flux-specific transition times offer unprecedented opportunity to refine the process representations in models even further (Baldocchi et al., 2001;Falge et al., 2002;Noormets, 2009;Wofsy et al., 1993). The remainder of this paper is organized to the dataset description, the summary of estimating the seasonality metrics



from idealized seasonal curves of different fluxes, description of the model performance, uncertainties of the flux seasonality metrics, and conclusions.

## 2 Data

FLUXNET is a global network of regional networks of eddy covariance sites that measure the exchange of $CO_2$, water vapor, and energy between vegetation and the atmosphere (Baldocchi et al., 2001;Baldocchi, 2008). Recently, harmonized data processing protocols have been developed (Pastorello et al., in preparation), and the growing global coverage of these observations has become the de-facto ground-truthing tool for both mechanistic ecosystem models as well as global planetary circulation models (Baldocchi, 2003;Baldocchi, 2020). The data include continuous (i.e. gap-filled) measurements of net ecosystem exchange of $CO_2$ (NEE), latent and sensible heat fluxes (LE and H), and microclimate data (air temperature, humidity, wind speed and direction, solar radiation, soil temperature, and soil water content), all at a 30-minute time step. Estimates of canopy photosynthesis and ecosystem respiration, derived from the data using an empirical model, are also typically available. The data undergo quality assurance, and missing half-hourly averages are filled using standardized methods to provide continuous data records (Papale et al., 2006). The current study uses FLUXNET2015 Dataset (https://fluxnet.fluxdata.org/), which includes over 200 sites and around 1500 site-years of data (Figure 1). Of all the sites, 67 sites have no less than 10 years' observations. The gap-filled 30-minute data series of fluxes and micrometeorological conditions was aggregated to daily totals. The example sites for each biome were selected based on the following boundary conditions: 1) distinct seasonality of all fluxes; 2) data coverage of observed or high-quality gap-filled data >75%. Therefore, the coverage of different fluxes is different, in which GPP has the highest coverage and H has the lowest coverage. Even

though the FLUXNET mainly distributed in the northern-hemisphere and temperate ecosystems, it still has high spatial and temporal representativeness (Yu et al., 2019).

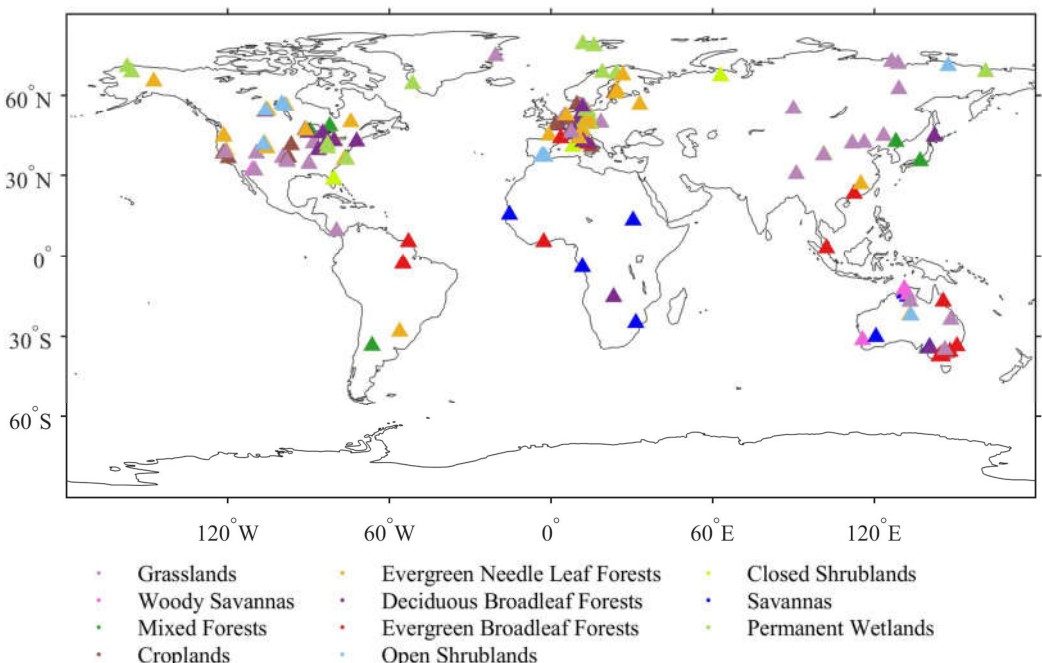

**Figure 1: Global distribution of eddy covariance flux sites included in this study; color coded by their International Global Biosphere**
**Programme (IGBP) biome classes.**

Although there is a broad agreement between different flux partitioning approaches (Moffat et al., 2007), and many approaches have converged recently, the current FLUXNET data product still includes a couple of alternative estimates of derived fluxes (RE, GPP). The latest interpretation of respiration fluxes, in particular, is that the night- and daytime estimates may represent different facets of "truth" (Keenan et al., 2019). Different researchers may choose different partitioning approaches for different
purposes. Hence, the phenological metrics dataset includes metrics for all biogeochemical fluxes reported in the FLUXNET dataset. Debating the strengths of different partitioning approaches, and indeed, the quality of the underlying dataset is beyond the scope of this study. We assume that the data reported has passed certain thresholds, even though some additional screening has been necessary, and not all site-years are of sufficient coverage and quality to estimate the flux seasonality metrics. The metrics are reported both in absolute flux scale (to allow comparisons against commonly reported values) and in relative,
normalized scale (to allow comparisons of development rates among different fluxes).



## 3 Method

### 3.1 Phenology metrics and uncertainties from flux observation

The seasonal dynamics of the ecosystem fluxes generally have five distinctive phases, which results from the interaction between the inherent biological and ecological processes and the changes in environmental conditions and reflects the

unique functioning of plant community at different stages of the growing season (Gu et al., 2009). The five phases are 1) Pre-phase, baseline dormant season flux, before leaf development; 2) Flux development period, a rapid increase in flux rate, concurrent or immediately following leaf emergence and expansion; 3) Peak flux period, a relatively steady stage in the middle of the growing season; 4) Flux recession period, a rapidly declining stage to the baseline; 5) Termination phase, the onset of a new dormant season, following leaf senescence and abscission.

Here, we fit the single-pass double-logistic model as first described by Gu et al. (2009) to fit the flux time series. This function exhibits broad structural flexibility and robust convergence, both of which are important for automated processing. The temporal variation in eddy-flux data for an entire growing cycle can be modeled using the function:

$$F_m(t) = f_0 + \frac{a_1}{\left(1+e^{-b_1(t-t_1)}\right)^{c_1}} - \frac{a_2}{\left(1+e^{-b_2(t-t_2)}\right)^{c_2}} \tag{1}$$

where $F_m$ is the flux value in a given day of year (DOY) $t$, $f_0$ is the dormant season base flux, and $a_1$ and $a_2$ are parameters

about the flux magnitude. Parameters $b_1$, $b_2$, $c_1$ and $c_2$ are related with the transitions or curvature parameters. The model was fit to daily integrated fluxes, following iterative procedures:

    a)    Fit Equation (1) to the flux time series, and calculate a predicted value for each DOY.

    b)    For each point in the time series, compute the ratio of the observed to predicted flux.

    c)    Conduct the Grubb's test to identify outliers in the obtained ratios.

115       d)    If an outlier is detected, remove this outlier and go to step c.

    e)    If no outliers are found, remove the data points whose ratios are more than one standard deviation (1s) below the mean ratio.

    f)    Fit Equation (1) to the time series of the daily flux measurements.

The DOYs at which the fitted logistic curve showed characteristic curvature changes were identified with the formula shown

in Table 1 derived analytically from the seven parameters of Equation (1) corresponding to the minimum and maximum values of the first and second derivatives. The first derivative of Equation (1) is given by:

$$F_m{}'(t) = \frac{a_1 c_1 e^{-(b_1(t-t_1))}}{b_1 \left(1+e^{-b_1(t-t_1)}\right)^{c_1+1}} - \frac{a_2 c_2 e^{-(b_2(t-t_2))}}{b_2 \left(1+e^{-b_2(t-t_2)}\right)^{c_2+1}} \tag{2}$$

The first derivative can be considered as the rate of change of the flux. The maximum of the first derivative occurs early and the minimum late in the growing season (Figure 2). The day on which the maximal growth rate of each flux occurs "Midpoint of flux development period" (DOY$_{MFD}$; point B),  the day on which the minimal growth rate occurs is termed as "Midpoint of



flux recession period" (DOY$_{MFR}$; point E), and the interval between these two transition dates is termed as "Length of Flux Midpoint" (L$_{FM}$=E-B).

The second derivative of Equation (1) is given by:

$$F_m''(t) = \frac{a_2 c_2 e^{-(b_2(t-t_2))}}{b_2^2(1+e^{-(b_2(t-t_2))})^{c_2+1}} - \frac{a_1 c_1 e^{-(b_1(t-t_1))}}{b_1^2(1+e^{-(b_1(t-t_1))})^{c_1+1}} + \frac{a_1 c_1 e^{-(2b_1(t-t_1)(c_1+1))}}{b_1^2(1+e^{-(b_1(t-t_1))})^{c_1+2}} - \frac{a_2 c_2 e^{-(2b_2(t-t_2)*(c_2+1))}}{b_2^2(1+e^{-(b_2(t-t_2))})^{c_2+2}} \tag{3}$$

The second derivative can be considered as the rate of the growth rate of the flux. The spring and fall maxima of the second derivative mark "Start of Flux development period" (DOY$_{SFD}$; point A) and "End of the Flux recession period" (DOY$_{EFR}$; point F), whereas the minima mark the "End of the flux development period (DOY$_{EFD}$)\Start of the peak flux period (DOY$_{SPF}$)" (point C) and "End of the peak flux period (DOY$_{EFP}$)\Start of the flux recession period (DOY$_{SFR}$)" (point D). Periods between AC, CD and DF mark the length of flux development, peak flux, and flux recession periods (L$_{FD}$, L$_{PF}$ and L$_{FR}$, respectively).

We also calculated the peak flux (Fmax), date of peak flux (DOY$_{Fmax}$), the rate of the flux development period (R$_{FD}$) and the rate of the flux recession period (R$_{FR}$). Period AF is the length of active season (L$_{AS}$).

The fitted daily fluxes observations were normalized to 0-1 and then the rates of change were calculated, which can be used for comparison of development rates among different fluxes.

All the seasonality metrics were summarized in Table 1.

**Table 1: Seasonality metrics estimated for biogeochemical fluxes gross primary production, ecosystem respiration, latent heat and sensible heat.**

| Metric type | Abbreviation | Name of metric | Figure 2 label | unit |
|---|---|---|---|---|
| Transition dates | DOY$_{MFD}$ | Midpoint of flux development | B | DOY |
| | DOY$_{MFR}$ | Midpoint of flux recession | E | DOY |
| | DOY$_{SFD}$ | Start of flux development | A | DOY |
| | DOY$_{EFD}$=DOY$_{SPF}$ | End of flux development/Start of peak flux period | C | DOY |
| | DOY$_{SFR}$=DOY$_{EPF}$ | End of peak flux period/Start of flux recession period | D | DOY |
| | DOY$_{EFR}$ | End of flux recession period | F | DOY |
| | DOY$_{Fmax}$ | Date of peak flux | G | DOY |
| Phase durations | L$_{MF}$ | Length of flux midpoint | BE | days |
| | L$_{FD}$ | Length of flux development period | AC | days |
| | L$_{PF}$ | Length of peak flux period | CD | days |
| | L$_{FR}$ | Length of flux recession period | DF | days |
| | L$_{AS}$ | Length of active season | AF | days |
| Peak Value | Fmax | Peak flux value | - | - |
| Rates of change | R$_{FD}$ | Rate of flux development | - | - |
| | R$_{FR}$ | Rate of flux recession | - | - |








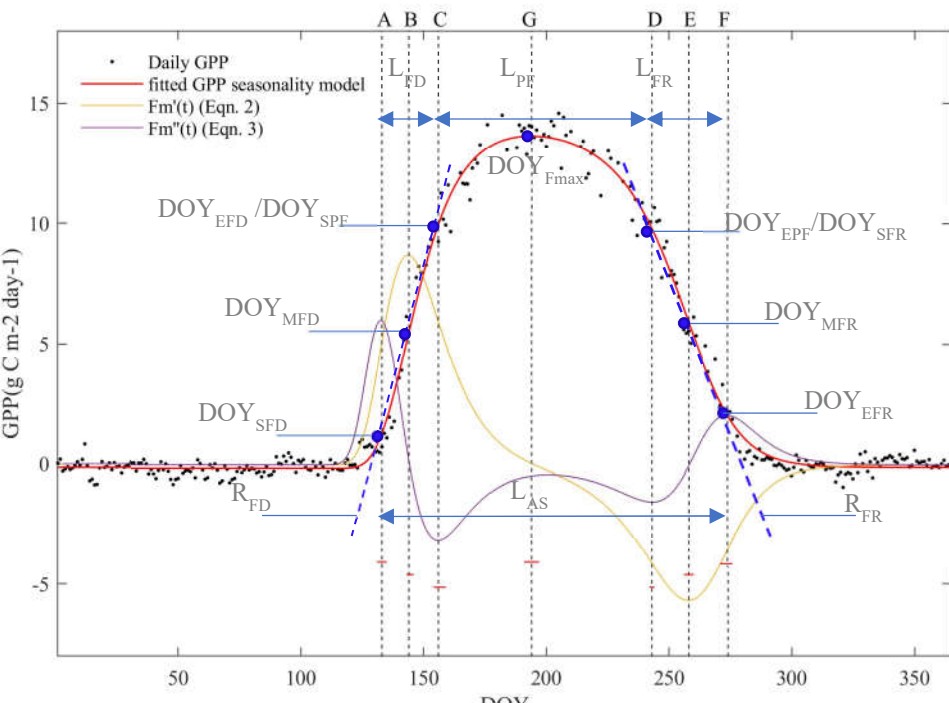

**Figure 2: An example of seasonal dynamics of gross primary productivity (GPP), and metrics of transition points of the different phases derived from the extremes of the first ($F_m'(t)$) and second ($F_m''(t)$) derivatives of the fitted logistic function Eq. (1). For**
**visual clarity, the scales of the first and second derivatives are enhanced 20-fold and 200-fold, respectively. The red line indicates the double-logistic model (Eqn. 1) fitted to the observed flux time series (black dots). The slope of the blue dash lines indicated the rate of change during the flux development/ recession period. The phenological transition points are marked with the vertical dashed lines, and the bootstrap estimates of 95% confidence intervals of these metrics are indicated with the horizontal red lines about each vertical line.**

**3.2 Evaluation of the quality**

**3.2.1 Model fit statistics**

The fit of Equation (1) to flux time series was characterized through the coefficient of determination ($R^2$), root mean square error (RMSE), empirical BIAS, and agreement index (AI).

The $R^2$ value of a regression is a measure of the portion of the variance of the dependent variable accounted for by the
explanatory variables, and characterizes the goodness of fit of the fitted model,

$$R^2 = \frac{\sum_{t=1}^{n}(F_o(t)-\overline{F_o})^2 - \sum_{t=1}^{n}(F_o(t)-F_m(t))^2}{\sum_{t=1}^{n}(F_o(t)-\overline{F_o})^2} \tag{4}$$

where $F_m$ and $F_o$ were the predicted and observed values, respectively, $\overline{F_o}$ is the mean value of the observations, and n is the sample size or the number of days in the year.





The RMSE was estimated as the square root of the mean value of the squared residuals:

$$\text{RMSE} = \sqrt{\frac{\sum_{t=1}^{n}(F_o(t)-F_m(t))^2}{n}} \tag{5}$$

The BIAS was calculated as the mean value of the model's residuals:

$$\text{BIAS} = \frac{\sum_{t=1}^{n}(F_o(t)-F_m(t))}{n} \tag{6}$$

Agreement index (AI; Willmott, 2013) provides a measure of the relative error in model estimates, combining the information contained in the correlation coefficient (R) and RMSE, and is popular in model assessments and calibration (Gu et al.,
2002;Zhou et al., 2016). It is calculated as:

$$\text{AI} = 100 - 100\frac{\sum_{t=1}^{n}(F_o(t)-F_m(t))^2}{\sum_{t=1}^{n}(|(F_m(t)-\overline{F_o}|+|F_o(t)-\overline{F_o}|)^2} \tag{7}$$

AI is dimensionless and ranges from 0 (complete disagreement) to 100 (perfect fit). The AI is also sensitive to differences between observed and modeled means (Willmott, 2013). Thus, the AI is well suited for comparing model fits across different biomes and climates.

**3.2.2 Uncertainties calculation**

The uncertainties in the flux seasonality metrics estimates arise from two sources: (i) the day-to-day variability of fluxes, particularly during the transition periods, that affect the overall goodness of fit of Equation (1) (see section 3.2.1), and (ii) the consistency of change in climatological drivers during the transition periods that can manifest as early or late cold or warm spells providing conflicting signals to plant development and can affect specific metrics without affecting others. The overall
model fit statistics can be used to identify the suitability of different data sources for flux seasonality assessment, but they are not good indicators of the confidence in specific seasonality metrics. Assessing the quality of the underlying flux data is beyond the scope of the current study, and all reported flux values are assumed "true" and the best possible estimates. The uncertainties in the seasonality metrics were estimated similar to (Elmore et al., 2012), using Monte Carlo bootstrapping (Efron, 1979). Bootstrapping is a statistical procedure that resamples a single dataset to create many simulated samples. The advantage of the
bootstrapping is that parameters can be estimated without assumptions about the normal distribution and using also small sample size. The distribution of parameter estimates for these bootstrap models provides valuable information about parameter uncertainty and correlation that is free of assumptions about the underlying data distributions. In this study, random uniform sampling with replacement was conducted for 500 times for each site-year, and the seasonality metrics were estimated for each iteration of the bootstrapped dataset. The 5[th] and 95[th] percentiles of the 500 bootstrapped phenology metrics estimates were
taken as the confidence interval of the mean estimated from the original dataset (Elmore et al., 2012;Klosterman et al., 2014).



## 4 Results

### 4.1 Model fit statistics

The double logistic model (Equation (1)) captured the temporal dynamics of widely divergent flux time series (Figure 3). Although the model fit statistics do not directly translate to the quality of the seasonality metrics estimates (see section 3.2.2), the general fit statistics deserve a brief review. Table 2 shows the overall performance of the fitted model for the different fluxes. The primary explanatory variable behind the fit statistics, as well as the differences between different fluxes, was the range of day-to-day variability in the flux time series. For example, H was generally more variable than LE, both of which are much more variable than RE and GPP, which resulted in lower fit statistics for H and LE than for GPP and RE (Table 2, Figure 3). The model fit statistics reported in Table 2 were largely consistent in ranking the goodness of fit among biomes. Biomes

with well express seasonal flux magnitude differences (mixed forest, deciduous broadleaf forest, evergreen needleleaf forest, croplands) exhibited consistently higher fit statistics than biomes with weak seasonality of fluxes (CSH and EBF). The latter also exhibited relatively greater day-to-day variability of fluxes, resulting in lower fit statistics. For GPP, the fit statistics were practically indistinguishable for time series partitioned based on daytime and nighttime partitioning methods (GPP-DT: AI = 98.654, $R^2$ = 0.951; GPP-NT: AI = 98.483; $R^2$ = 0.946), whereas for RE the nighttime partitioning method showed marginally

better fit than the daytime method (AI = 98.654 and $R^2$ = 0.935 versus AI = 96.614 and $R^2$ = 0.885).

**Table 2: Model fit statistics to different flux time series in different biomes.**

| Type | $R^2$ | | | | | | RMSE | | | | | | Bias | | | | | | AI | | | | | |
|---|---|---|---|---|---|---|---|---|---|---|---|---|---|---|---|---|---|---|---|---|---|---|---|---|
| | RE | | GPP | | H | LE | RE | | GPP | | H | LE | RE | | GPP | | H | LE | RE | | GPP | | H | LE |
| | DT | NT | DT | NT | CORR | CORR | DT | NT | DT | NT | CORR | CORR | DT | NT | DT | NT | CORR | CORR | DT | NT | DT | NT | CORR | CORR |
| All | 0.885 | 0.935 | 0.951 | 0.946 | 0.792 | 0.912 | 0.511 | 0.37 | 0.517 | 0.562 | 12.749 | 8.113 | 0 | 0 | 0.001 | 0 | 0.002 | 0.01 | 96.614 | 98.134 | 98.654 | 98.483 | 93.34 | 97.528 |
| DBF | 0.886 | 0.949 | 0.981 | 0.98 | 0.695 | 0.953 | 0.564 | 0.368 | 0.528 | 0.554 | 14.395 | 8.136 | 0 | 0 | 0.001 | 0 | 0.004 | 0.007 | 96.682 | 98.622 | 99.512 | 99.487 | 89.532 | 98.764 |
| EBF | 0.738 | 0.859 | 0.815 | 0.815 | 0.854 | 0.858 | 0.826 | 0.629 | 0.787 | 0.879 | 14.129 | 9.197 | -0.002 | 0 | 0 | -0.003 | 0.006 | 0.017 | 91.258 | 95.837 | 94.456 | 94.393 | 95.654 | 95.823 |
| ENF | 0.894 | 0.951 | 0.954 | 0.955 | 0.803 | 0.917 | 0.511 | 0.343 | 0.536 | 0.552 | 15.468 | 7.961 | -0.001 | 0.001 | 0.001 | 0 | 0.001 | 0.011 | 96.917 | 98.637 | 98.779 | 98.802 | 94.077 | 97.744 |
| CRO | 0.934 | 0.953 | 0.961 | 0.957 | 0.761 | 0.905 | 0.455 | 0.369 | 0.574 | 0.633 | 10.982 | 10.25 | 0.001 | 0 | 0.001 | 0 | -0.003 | 0.028 | 98.22 | 98.735 | 98.92 | 98.816 | 92.026 | 97.369 |
| OSH | 0.856 | 0.866 | 0.914 | 0.887 | 0.912 | 0.872 | 0.151 | 0.095 | 0.136 | 0.174 | 10.091 | 4.835 | 0 | 0 | 0 | 0 | 0.017 | 0.004 | 95.448 | 95.626 | 97.421 | 96.639 | 97.61 | 96.299 |
| CSH | 0.79 | 0.885 | 0.847 | 0.864 | 0.869 | 0.745 | 0.425 | 0.283 | 0.458 | 0.459 | 15.26 | 8.344 | 0 | 0 | 0 | 0 | 0.001 | 0 | 93.513 | 96.688 | 95.605 | 96.113 | 96.037 | 91.597 |
| GRA | 0.911 | 0.941 | 0.954 | 0.939 | 0.804 | 0.912 | 0.495 | 0.427 | 0.518 | 0.598 | 8.331 | 8.123 | 0 | 0 | 0.001 | 0 | 0 | 0.01 | 97.526 | 98.296 | 98.779 | 98.306 | 93.568 | 97.53 |
| MF | 0.854 | 0.911 | 0.96 | 0.948 | 0.747 | 0.944 | 0.648 | 0.426 | 0.584 | 0.667 | 15.037 | 6.711 | 0.001 | 0.001 | 0.003 | 0 | 0.003 | 0.018 | 97.323 | | 98.962 | 98.606 | 92.209 | 98.538 |
| SAV | 0.862 | 0.901 | 0.905 | 0.87 | 0.78 | 0.897 | 0.445 | 0.351 | 0.385 | 0.508 | 10.392 | 7.086 | 0 | 0 | 0 | 0 | 0.03 | 0.001 | 95.778 | 97.042 | 97.198 | 95.872 | 92.149 | 96.966 |
| WSA | 0.846 | 0.902 | 0.926 | 0.919 | 0.934 | 0.922 | 0.456 | 0.313 | 0.387 | 0.402 | 10.144 | 5.749 | 0 | 0 | 0 | 0 | -0.011 | 0.007 | 95.459 | 97.197 | 98.004 | 97.796 | 98.213 | 97.898 |
| WET | 0.923 | 0.949 | 0.979 | 0.975 | 0.755 | 0.909 | 0.347 | 0.273 | 0.303 | 0.336 | 10.27 | 7.679 | 0 | 0 | 0.001 | 0 | 0.011 | -0.059 | 97.863 | 98.554 | 99.456 | 99.336 | 91.967 | 97.038 |

$R^2$: coefficient of determination; RMSE: root mean square error; AI: agreement index; GPP: gross primary production; RE: ecosystem respiration; H: sensible heat flux; LE: latent heat flux; DT: Daytime partitioning method; NT: Nighttime partitioning method; CORR: corrected H_F_MDS or LE_F_MDS. DBF: deciduous broadleaf forests; EBF: evergreen broadleaf forests; ENF: evergreen needle leaf

forests; CRO: croplands; OSH: open shrublands; CSH: closed shrublands; GRA: grasslands; MF: mixed forests; SAV: Savannas; WSA: woody Savannas; WET: permanent wetlands.







**Figure 3: Examples of the seasonal dynamics of different fluxes for 10 sites representative of different biomes. One biome, open**
**shrubland was left off because of space limitations on a single page. The red line indicates the double-logistic model (Equation 1)**
**fitted to the observed flux time series (black dots). The phenological transition points are marked with the vertical dashed lines, and**
**the bootstrap estimates of 95% confidence intervals of these metrics are indicated with the horizontal red lines about each vertical**
**line.**

## 4.2 Uncertainties of seasonality metrics

The uncertainties of individual flux seasonality metrics (Table 1), estimated as the 5th and 95th percentiles of 500 Monte Carlo
bootstrapping samples ranged from about a week to several weeks, and the uncertainties of phase durations tended to generally
be greater than those of individual transition dates. Generally, the uncertainties were the lowest for the phenology metrics of
GPP, and highest for H (shown as horizontal red lines on Figure 3). The average uncertainties of transition dates ranged from
6-11 days for GPP, 8 to 14 days for RE, 10 to 15 days for LE, and 15 to 23 days for H. The average uncertainties of duration
length ranged from 12-20 days for GPP, 14 to 23 days for RE,16 to 25 days for LE, and 23 to 32 days for H.

    For all fluxes, deciduous broadleaf forest always showed the lowest uncertainties among all biomes. Uncertainties of flux
development midpoints were always lower than those of start- and endpoints. Meanwhile, the uncertainties of duration dates
are larger than those of transition dates, indicating the compounding effect of the uncertainties of the start and end dates of the
active season. The length of the dormant season also affects the uncertainties: the longer the dormant season, the lower the
uncertainties.

## 4.3 Alternative data sources

### 4.3.1 Daily peak versus daily total fluxes

    Even with standardized data sources like FLUXNET2015 Dataset, some user discretion of data aggregation remains, which
may affect the reproducibility of different analyses. We mentioned earlier that FLUXNET2015 Dataset contains GPP and RE
estimates from different gap-filling algorithms. In addition, the daily time series can be assembled as either daily integrated
fluxes (more common) or as daily peak fluxes, as proposed by Gu et al. (2009), reasoning that daily peaks may be less sensitive
to cloudiness and thus better capture the seasonal dynamics of flux capacity. To test this line of argumentation, and identify
the best representation of flux seasonality, we started the current study by comparing the model fit statistics for either of these
data types. Except for H, all other fluxes had higher model fit statistics with daily integrated than daily peak fluxes (Figure 4).
Although only $R^2$ are shown, all other fit statistics confirmed the same pattern (Table 2). Therefore, we adopted the daily
integrated fluxes for the following phenology metrics dataset generation. Although the day- and nighttime partitioning methods
yielded different flux estimates, and different model fits, choosing the "best" between them is not appropriate at this point.
Both partitioning methods have their uses, and their respective merits have not been conclusively proven. The differences
between the seasonality metrics of each dataset will be discussed in section 4.2.3.



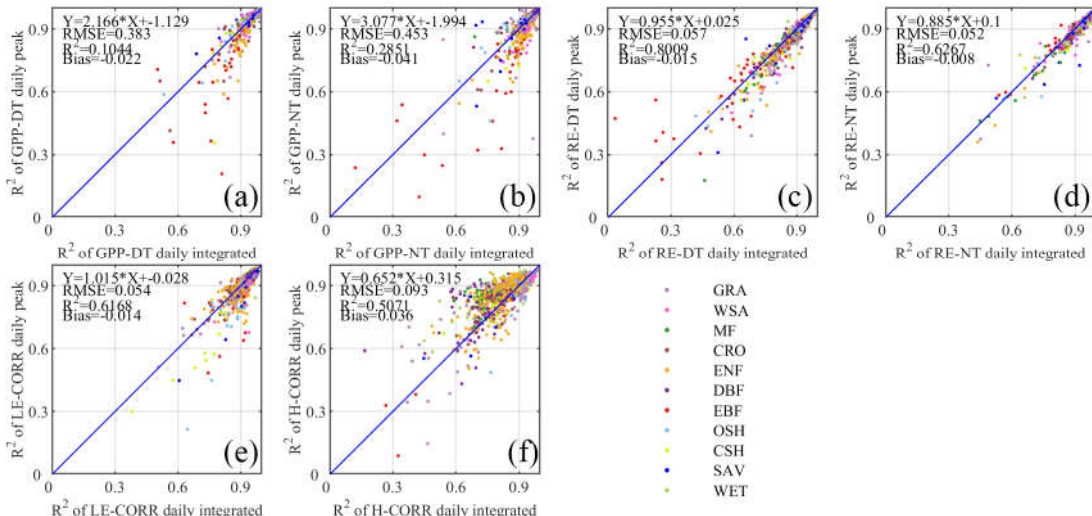

**Figure 4: Comparison of the coefficients of determination ($R^2$) of fitted seasonality curves between daily integrated fluxes (x-axis) and daily peak fluxes (y-axis) for GPP (a, b), RE (c, d), LE (e) and H (f). Both daytime (DT) and nighttime (NT) flux partitioning models are shown for GPP and RE.**

Although daily flux totals were identified as the preferred scalar for deriving seasonality metrics from, we will report here the differences between the metrics estimated from the daily peak and daily total ecosystem respiration (Figure 5). This can be viewed as the minimum methodological uncertainty in a "best-case scenario" in the sense that the correlation between the two sets of metrics was much higher (average $R^2 = 0.82$; Figure 5) than between other sources of variability (e.g. the daytime and nighttime partitioning models resulted in seasonality metrics with an average $R^2 = 0.29$; Figure 7). The differences between the metric derived from daily integrated and peak flux values are generally smaller than the confidence intervals of individual estimates, except in WSA and SAV. The only metric with a distinct difference between the data types was annual peak flux, where the annual peak fluxes exhibited consistently greater values than daily integrals (last panel on Figure 5), as would be expected.

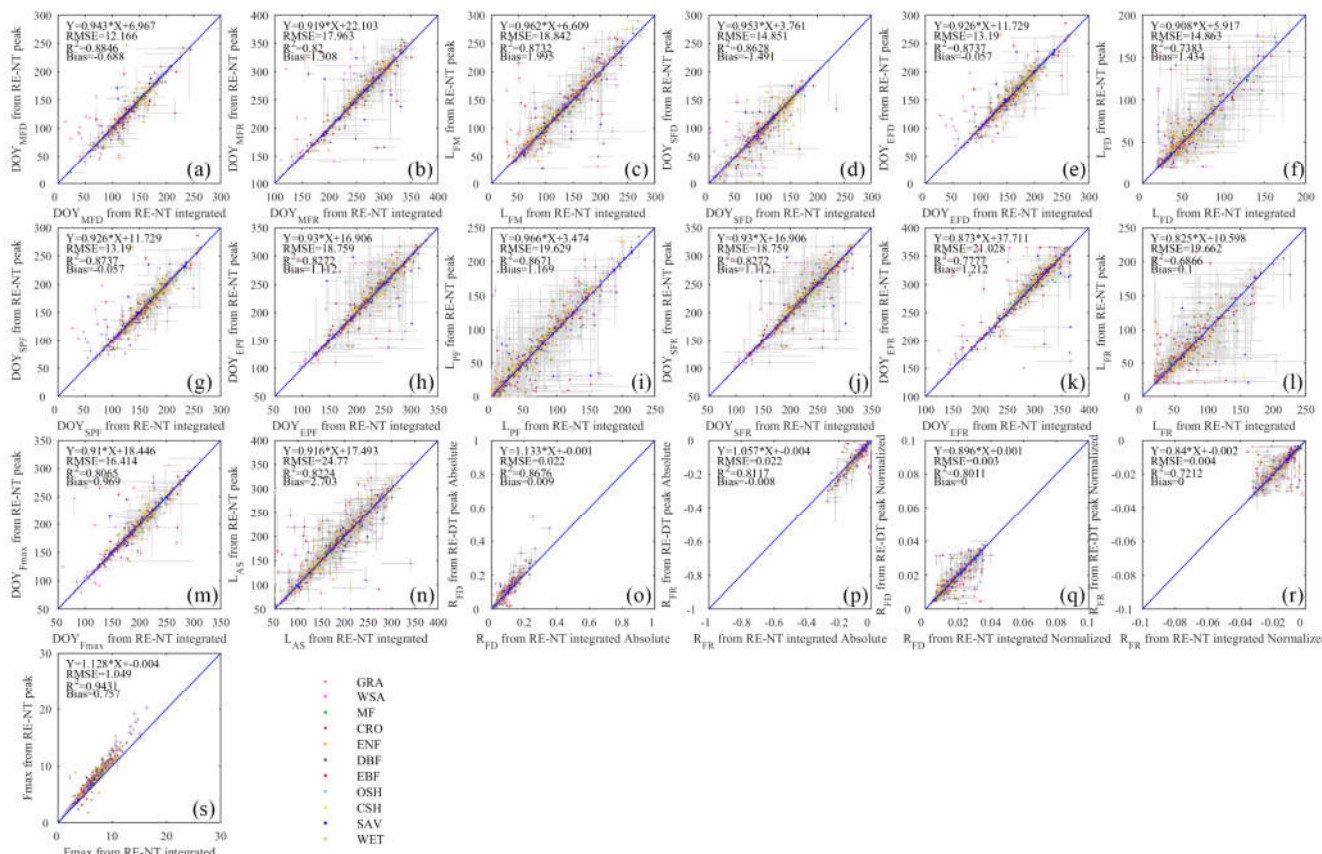

**Figure 5: The scatter plots of different phenology metrics from RE-NT daily integrated data and RE-NT daily peak data.**

### 4.3.2 Comparison of different partitioning methods

The significance of the assumptions made by partitioning methods to fill flux time series has been emphasized from the perspective of flux integrals (Kruijt et al., 2004). Here we show that the choice of the partitioning model can also affect the seasonality of daily integrated fluxes, and thus the seasonality metrics (Figure 6). We report here the differences between the daytime and nighttime models of flux partitioning as exemplified by the RE and GPP time series, but the lessons apply for all partitioning approaches. Most importantly, mixing of time series filled with different models should not be done.

#### 4.3.2.1 RE

The nighttime and daytime flux partitioning methods can yield similar or dissimilar daily RE, and sometimes even the seasonality can diverge significantly between them (e.g. US-PFa 2014 in Figure 6).

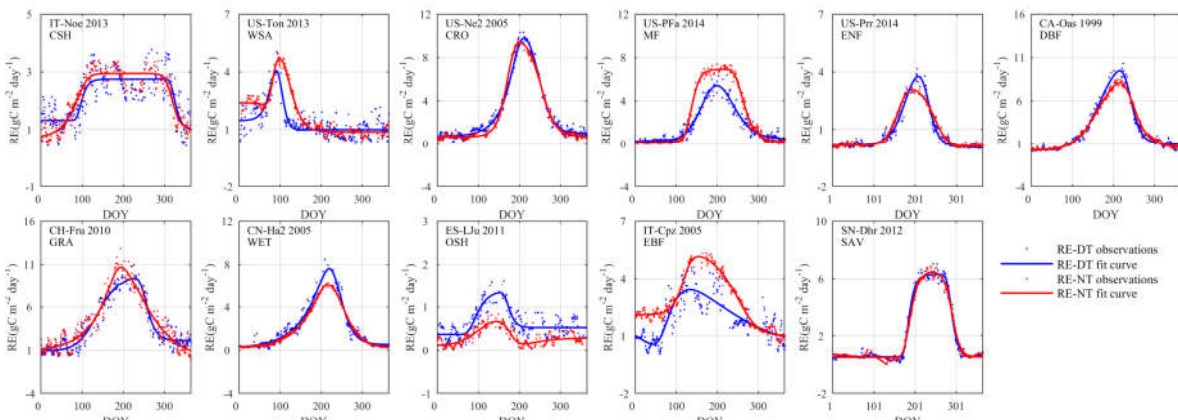

**Figure 6: Seasonal dynamics of RE from daytime and nighttime partitioning methods for 11 sites representative of different biomes.**
The differences in flux seasonality metrics based on the partitioning method (Figure 7) are sizeable, non-systematic, and
generally greater for season length metrics than transition date metrics (because the season length is determined by two
transition dates, each of which is subject to deviation among methods). Interestingly, the seasonality metrics of flux
development period (Figure 7a, d, e, f) are more consistent than those of the peak flux period (Figure 7g, h, i) and flux recession
period (Figure 7b, j, k, l). A more detailed analysis of the performance of the consistency of the different partitioning methods
is the subject of future studies, and we may find answers more from information pertaining more to the accuracy of the flux
estimates than the seasonality estimates.

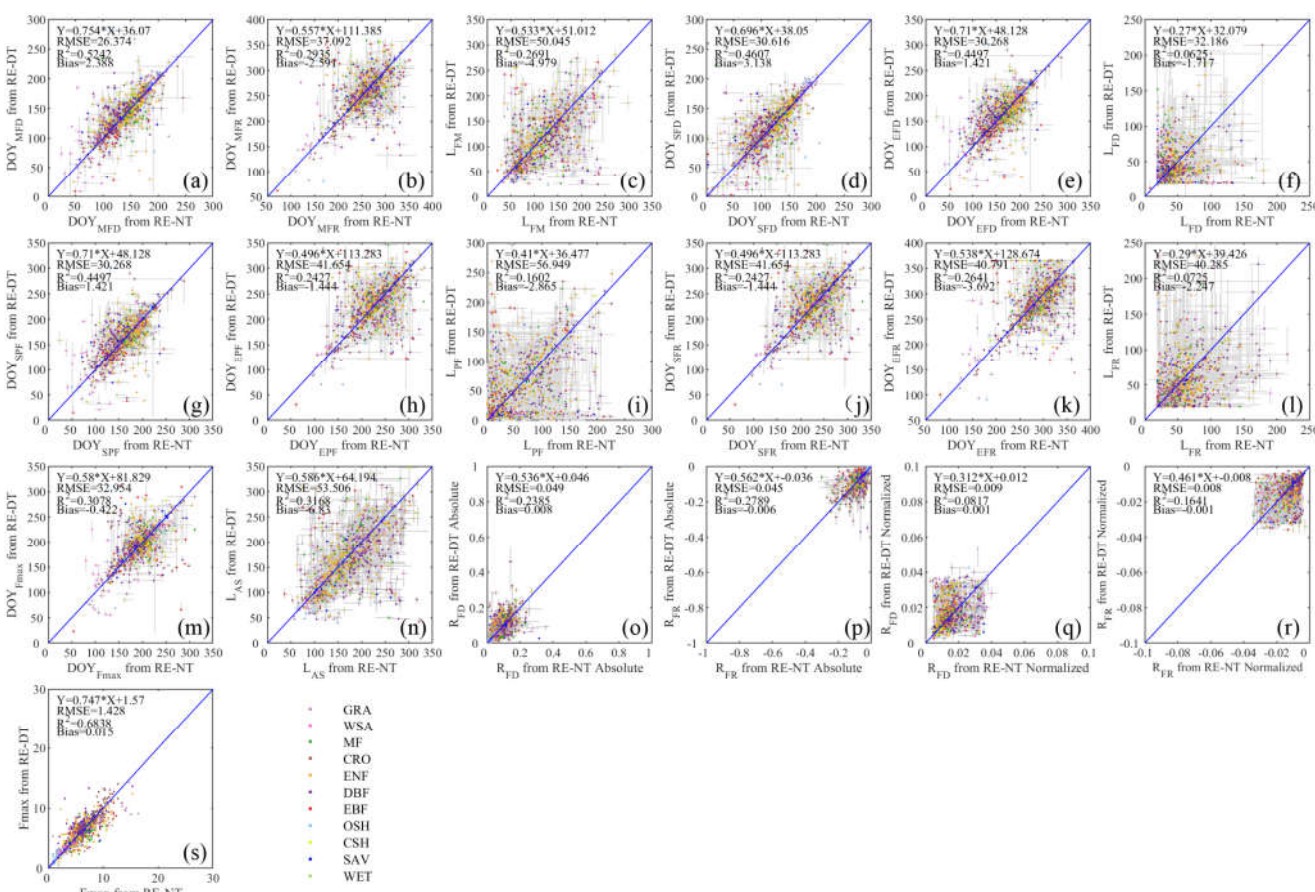

**Figure 7: The scatter plots of seasonality metrics from RE data using daytime and nighttime partitioning methods.**

### 4.3.2.2 GPP

The seasonality of GPP time series differed less between the two flux partitioning methods than did RE (Figure 8). It is also obvious that the seasonalities of RE and GPP for the same sites differed significantly in terms of the seasonal timing, symmetry, peak duration, and other aspects. A more detailed assessment of these differences is the subject of a forthcoming analysis (Yang and Noormets, In preparation).

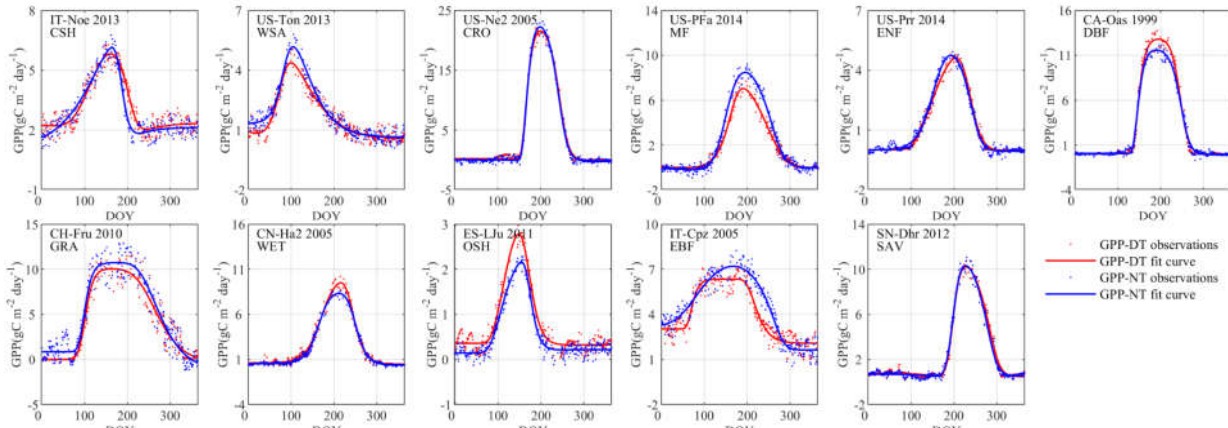


**Figure 8: Seasonal dynamics of GPP from daytime and nighttime partitioning method for 11 sites representative of different biomes.**

As suggested by the extent of overlap in the flux estimates on Figure 8, the seasonality metrics of GPP from the two partitioning methods were also more consistent compared to RE (Figure 9). The scatter was distributed around the 1:1 line and with little bias. However, similar to RE, the metrics of the flux development period of GPP were also more consistent among the

partitioning methods than those of peak flux period and flux recession period. Yet, the rate of flux development was more variable between the methods than the rate of flux recession (Figure 9o, p, q, r). And like for ER, the phase duration metrics of GPP were more variable than the transition dates.



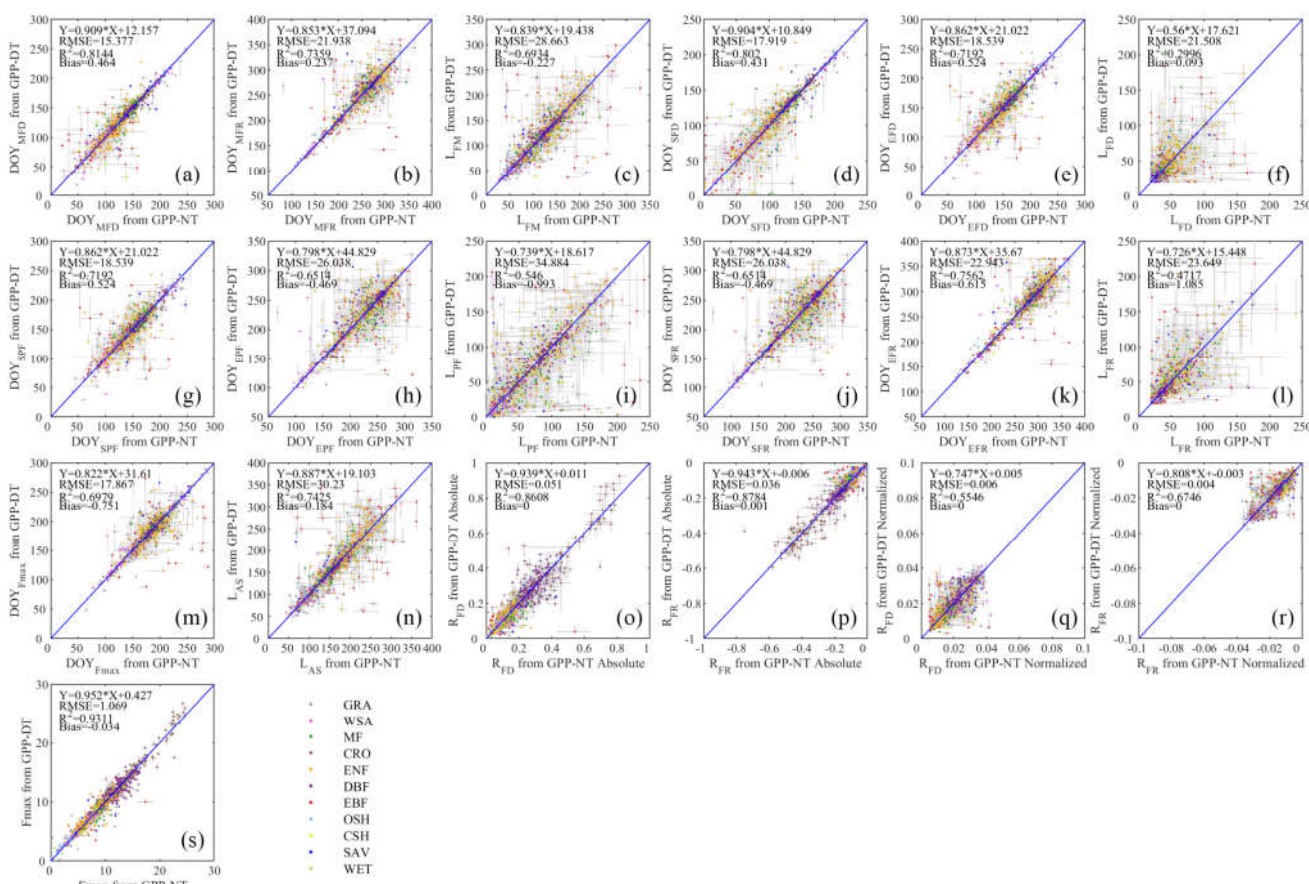

**Figure 9: The scatter plots of different phenology metrics from GPP data using daytime and nighttime partitioning methods.**

## 5 Significance

The Flux Seasonality Metrics Database (FSMD) presented here summarizes important metadata contained in the land surface biogeochemical fluxes, and is likely to allow novel insights to the functioning of the biosphere, and assist in the development and validation of novel functionality in Earth System Models.

Improving the predictive capabilities of ecosystem biogeochemistry models on interannual and decadal scales remains a current challenge, and variability in the seasonality of different fluxes has been recognized as a key uncertainty. Importantly, the seasonality of GPP in models is often forced to match observations with arbitrary coefficients (Straube et al., 2018), as the divergence of LE and GPP seasonality is not captured in common LAI-driven models (Wu et al., 2017;Restrepo-Coupe et al., 2017). By developing process-specific seasonality references, explicit validation of these fluxes becomes possible. In addition, the ability to discern shifts in seasonality from those in flux capacity or vegetation structure can also be important in correcting the attributions of observed changes.

A standardized flux seasonality metrics dataset can also support other seasonality assessment tools, like near-surface (e.g. Phenocam Network; Richardson et al., 2012) and remote optical sensors (satellites; Broich et al., 2015;White and Nemani, 2006;Ganguly et al., 2010;Gamon et al., 2016). However, as these methods purport to infer GPP from the greenness information, they are vulnerable to the same lags between leaf development, LE and GPP that have undermined model development mentioned above. In addition, pixel heterogeneity and clouds can reduce the potential of remote sensing approaches.

The FSMD was designed to capture and depict the seasonality of different ecosystem processes. FSMD will be updated within 6 months of each major release of FLUXNET database, makes it possible to quantify the differences and similarities between different ecosystem processes in their responses to changes in climatic conditions. Some potential applications of this dataset have been mentioned before, and there are likely many others. We anticipate that the FSMD will stimulate new research in global change and Earth science disciplines where land-atmosphere exchange dynamics play a central role.

## 6 Data availability

This database can be obtained at U.S. Department of Energy's (DOE) Environmental Systems Science Data Infrastructure for a Virtual Ecosystem (ESS-DIVE, https://data.ess-dive.lbl.gov/view/doi:10.15485/1602532; Yang and Noormets, 2020).

## Conflict of interest

The Authors declare no conflict of interest.

## Author Contributions

LY and AN designed the research; LY implemented the research and wrote the manuscript; AN edited and revised the manuscript.

## Acknowledgements

This work was supported, in part, by U.S. Department of Energy's Office of Science through Ameriflux Management Project (award # M1800063) and USDA McIntire-Stennis Cooperative Forestry Program (project # 121209). LY was supported by Texas A&M AgriLife Research Strategic Initiative Assistantship. We thank everybody who has contributed data and effort to producing the FLUXNET database.



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
