# Peer review of "Standardized flux seasonality metrics: A companion dataset for FLUXNET annual product"

_Earth System Science Data, 2020_

## Referee Comment (RC1) · Anonymous Referee #1 · 23 Jul 2020

This paper extracted a series of standardized flux seasonality metrics through identifying key transition points and phase durations of carbon, water, and energy fluxes from the FLUXNET 2015 Dataset of about 200 sites and 1500 site-years of data. These metrics are useful to understand the ecosystem processes and their responses to climate change. However, the dataset presented in this paper was not enough exciting and attractive to readers, because these metrics were derived from existing FLUXNET dataset and some of metrics were widely reported and used, such as phenological events from GPP and NEE. The other flaw is that the dataset is at site scale rather than at global scale. The work is lack of originality and effort to publish on the ESSD. The major concerns are as below. 1. The meaningful of these metrics was not presented

clearly in the Introduction section. Why the authors presented these metrics and how they are different with other existed metrics you mentioned in Line35-50. 2. Line 79. How high-quality gap-filled data was defined. Which variable do you use to select the high-quality data. How many sites and site-years data were used after selecting by boundary conditions. 3. The double-logistic model was used in this study. The method was useful for many of sites and land cover types. However, for some special site data, it may be not suitable and should be descripted more clearly. For example, how to define and extract these metrics for those sites with multi-peak seasonal dynamic, such as double-cropping CRO, and SAV, Mediterranean and tropical ecosystems with complicated climate conditions. 4. L315.The significant of these metrics should be strengthened. What are their usefulness and where they can be applied in. Please adding more details on the contribution of these metrics to discover the mechanism of carbon and water processes and their responses to climate and improve the calibration of the ecosystem models.

---

## Referee Comment (RC2) · Geoffrey Henebry (Referee) · 16 Jan 2021

The authors offer an important dataset to the ESS community. By processing FLUXNET data to characterize the seasonality of fluxes at more than 200 sites across the planet, the authors have provided a benchmark dataset comparable to the land surface phenology datasets produced from MODIS and VIIRS time series. Moreover, the authors have taken care to investigate the robustness of their estimates obtained from seasonality modeling through resampling statistics on the one hand and alternative flux partitioning techniques on the other. I expect that these data will find many willing users.

[Figure]

The narrative is concise and well-written, but I do have a few minor edits and comments.

line 9: is it really high accuracy or rather high precision?

lines 29-30: you may want to include here a highly relevant publication that appeared after you submitted this manuscript:

Weltzin, J.F., Betancourt, J.L., Cook, B.I., Crimmins, T.M., Enquist, C.A., Gerst, M.D., Gross, J.E., Henebry, G.M., Hufft, R.A., Kenney, M.A. and Kimball, J.S., 2020. Seasonality of biological and physical systems as indicators of climatic variation and change. Climatic Change, 163(4), pp.1755-1771.

line 49: "compared" is more accurate than "truthed" since these approaches are looking at related but distinct processes.

line 68: update "in prep"?

line 89: I think that "same phenomenon" is more precise description than "truth" here

line 104: of course, abscission occurs in only a subset of these land covers and sites

line 114: citation needed for Grubbs' Test

line 229: decode MDS (or is this related to FSMD?)

line 270: This what? It is not clear to what the isolated relative pronoun points.

line 304: update "in prep"?

line 316: this use of "metadata" is odd and misleading. Try instead "latent features".

line 327: omit "satellites"

with the archived data, the docx file is named "Instrucations" and should be in pdf rather than Word.

---

## Editor Comment (EC1) · David Carlson (Editor) · 17 Jan 2021

The two reviews present divergent assessments: one quite positive, the other quiet negative. Please pay careful attention to both sets of comments in your response.

---

## Author Comment (AC1) · 24 Jan 2021

Referee #2

The authors offer an important dataset to the ESS community. By processing FLUXNET data to characterize the seasonality of fluxes at more than 200 sites across the planet, the authors have provided a benchmark dataset comparable to the land surface phenology datasets produced from MODIS and VIIRS time series. Moreover, the authors have taken care to investigate the robustness of their estimates obtained from seasonality modeling through resampling statistics on the one hand and alternative flux partitioning techniques on the other. I expect that these data will find many

willing users.

Response: Thank you very much for your positive and constructive comments. Below, we address every comment carefully and explain the corresponding changes in the manuscript.

The narrative is concise and well-written, but I do have a few minor edits and comments.

line 9: is it really high accuracy or rather high precision?

Reply: Thank you very much for your question. What we want to present is the seasonality of ecosystem processes can be decomposed to identify key transition points and phase durations with high accuracy because of high quality of eddy covariance data. So, we prefer to keep the original writing 'with high accuracy'. The precision is, to a degree, indicated by the confidence intervals of the parameter estimates, and is, in fact, quite low.

lines 29-30: you may want to include here a highly relevant publication that appeared after you submitted this manuscript: Weltzin, J.F., Betancourt, J.L., Cook, B.I., Crimmins, T.M., Enquist, C.A., Gerst, M.D., Gross, J.E., Henebry, G.M., Hufft, R.A., Kenney, M.A. and Kimball, J.S., 2020. Seasonality of biological and physical systems as indicators of climatic variation and change. Climatic Change, 163(4), pp.1755-1771.

Reply: Thank you for bringing this publication to our attention. We have included this citation as suggested.

line 49: "compared" is more accurate than "truthed" since these approaches are looking at related but distinct processes.

Reply: changed as suggested.

line 68: update "in prep"?

Reply: This paper has since been published, and the reference is updated.

[Figure]

line 89: I think that "same phenomenon" is more precise description than "truth" here

Reply: changed as suggested.

line 104: of course, abscission occurs in only a subset of these land covers and sites

Reply: Thank you for your comment. In this research, we only consider sites with 1) distinct seasonality of all fluxes; 2) data coverage of observed and high-quality gap-filled data >75% (defined by variable-specific data quality flags in the FLUXNET database (Reichstein et al., 2005)) (Line 78-80). Further, Sites with non-standard flux seasonality, where the R2 of model fit was below 0.75, were filtered out during general model fit assessment (Line 105-106).

line 114: citation needed for Grubbs' Test

Reply: citation has been added (Grubbs, F. E.: Procedures for Detecting Outlying Observations in Samples, Technometrics, 11, 1-21, https://doi.org/10.1080/00401706.1969.10490657, 1969.)

line 229: decode MDS (or is this related to FSMD?)

Reply: It is spelled out as Marginal Distribution Sampling, which is a version of look-up table used in gapfilling of fluxes. We have defined it in the manuscript, too.

line 270: This what? It is not clear to what the isolated relative pronoun points.

Reply: We are sorry we did not express clearly. Here, what we want to talk it the performance of seasonality metrics extracted from daily peak ecosystem respiration data and daily integrated ecosystem respiration data. To avoidambiguity, we have changed it to 'we will report here the differences between the metrics estimated from the daily peak ecosystem respiration and daily integrated ecosystem respiration data'.

line 304: update "in prep"?

Reply: Thank you for the attention to our further analysis work. We have finished it and

submitted to the journal Global Change Biology.

line 316: this use of "metadata" is odd and misleading. Try instead "latent features".

Reply: changed as suggested.

line 327: omit "satellites" with the archived data, the docx file is named "Instrucations" and should be in pdf rather than Word.

Reply: Both changes made as suggested.

---

## Author Comment (AC2) · 24 Jan 2021

This paper extracted a series of standardized flux seasonality metrics through identifying key transition points and phase durations of carbon, water, and energy fluxes from the FLUXNET 2015 Dataset of about 200 sites and 1500 site-years of data. These metrics are useful to understand the ecosystem processes and their responses to climate change. However, the dataset presented in this paper was not enough exciting and attractive to readers, because these metrics were derived from existing FLUXNET dataset and some of metrics were widely reported and used, such as phenological

events from GPP and NEE. The other flaw is that the dataset is at site scale rather than at global scale. The work is lack of originality and effort to publish on the ESSD.

Response: Thank you for this perspective. It is true that phenological transition dates have been calculated for and analyzed based on both land- and space-based observations. However, to date there isn't a consistent terminology to describe the transitions, nor a comprehensive framework for deriving these metrics for different sites. Moreover, only the seasonality of GPP has been explored at any length, whereas the other biogeochemical fluxes have not been analyzed since the second author's book chapter over a decade ago. Since then, the FLUXNET database has grown exponentially, and has recently converged to globally harmonized data processing and sharing. The complementary dataset presented here is therefore timely and potentially valuable companion for this resource. We are also in communication with Ameriflux Data Team to have the workflow handed over and incorporated into their processing stream. Finally, this dataset can be used to validate other phenological products like land surface phenology products.

The major concerns are as below.

1. The meaningful of these metrics was not presented clearly in the Introduction section. Why the authors presented these metrics and how they are different with other existed metrics you mentioned in Line 35-50.

Reply: Thank you for the request to clarify. We have expanded this section, describing the relationship between ground based phenological observations, remote sensing seasonality metrics, and flux seasonality. What we report directly builds on the analyses of these earlier products, and we have cited the relevant publications. The novelty of the current study is in its universality of derivation of seasonality metrics for different fluxes, and quantifying aspects that previously have been addressed on a more ad-hoc basis (e.g. the change rate of fluxes during the spring and fall shoulder seasons). The framework is also open to incorporating other fitting models and strategies, as well as

applying the workflow to additional processes.

2. Line 79. How high-quality gap-filled data was defined. Which variable do you use to select the high-quality data. How many sites and site-years data were used after selecting by boundary conditions.

Reply: Thank you for this question. We have clarified in the paper that we used the variable-specific quality flags in the FLUXNET2015 data set. The flags indicate the fraction of daily coverage of observed and high-quality gapfilled records for the metric of selection. Specifically, we used FLUXNET2015 daily data product, which is integrated from half hourly data, and includes quality flags for all variables. The quality of individual 30-minute flux records is classified based on standard tests (originally based on Foken and Wichura 1996, reviewed by Mauder et al. 2008 and currently implemented in the harmonized data processing program ONEFlux (Pastorello et al. 2019), which is the data processing engine of the Ameriflux and ICOS workflows (Pastorello et al. 2020)), marking each record as: 0=measured, 1=good quality gapfilled, 2=medium quality gapfilled, 3=poor quality gapfilles. The quality of the daily aggregated values is expressed as a fraction of records with a quality flag value of 0 or 1. In this study, days with quality flag of 0.75 and higher were considered.

The final number of sites and site-years was as follows: for GPP: 169 sites and 1044 site-years; for RE: 173 sites and 1040 site-years; for LE: 134 sites and 834 site-years; for H: 128 sites and 800 site-years.

This information is summarized in the manuscript, lines 81-82: The final dataset included 169 sites and 1044 site-years for GPP, 173/1040 for RE, 134/834 for LE, and 128/800 for H.

3. The double-logistic model was used in this study. The method was useful for many of sites and land cover types. However, for some special site data, it may be not suitable and should be described more clearly. For example, how to define and extract these metrics for those sites with multi-peak seasonal dynamic, such as double-cropping

[Figure]

CRO, and SAV, Mediterranean and tropical ecosystems with complicated climate conditions.

Reply: Thank you for pointing this out. We have added a few sentences (lines 106-109) to explain how sites with non-standard flux seasonality were handled: Sites with non-standard flux seasonality, where the R2 of model fit was below 0.75, were filtered out during general model fit assessment. Most filtering related to poor quality gapfilling with obviously distorted seasonality, but 7 sites with multiple peaks during the same year were also excluded from the current study. Southern hemisphere sites were analyzed by shifting the calendar year cutoffs by 180 days. Future updates to this database are intended to include custom fits for sites with multiple peaks and phenological cycles across two different calendar years.

4. L315.The significant of these metrics should be strengthened. What are their usefulness and where they can be applied in. Please adding more details on the contribution of these metrics to discover the mechanism of carbon and water processes and their responses to climate and improve the calibration of the ecosystem models.

Reply: Thank you for this suggestion, but these aspects are covered in the Significance section of the manuscript. We mention several currently outstanding research themes that would benefit from the standardized flux seasonality dataset. Please see lines 322-339.

---

## Author Comment (AC3) · 24 Jan 2021

Dear Editor Carlson:

Hereby we submit the revised manuscript essd-2020-58 titled "Standardized flux seasonality metrics: A companion dataset for FLUXNET annual product". We have made the recommended changes, and admit that these clarifications improve the quality of this manuscript. The changes in response to each one is detailed in the Response files. Thank you for considering our work for "Earth System Science Data".

Sincerely yours,

Linqing Yang and Asko Noormets

---

## Author Response (AR2)

Thanks very much for your comments. We have made the suggested changes , and added some clarifying language. We admit that the uncertainty intervals on Figures 2 & 3 allowed misinterpretation. We have made revisions according to your comments and response point by point.

Please give special attention to Figure 2. In the version I see, some text remains hidden (lower left), text does not uniformly align with graphics (e.g. on the ascending side versus the descending side), etc.

Reply: We are sorry we did not realize the dislocation in last revision, and we have adjusted it to make everything all right.

[Figure]

**Figure 2: An example of the seasonal dynamics of gross primary productivity (GPP), and metrics of transition points of different phases derived from the extremes of the first ($F_m{'}(t)$) and second ($F_m{''}(t)$) derivatives of the fitted logistic function (Eq. 1). For visual clarity, the scales of the first and second derivatives are enhanced 20-fold and 200-fold, respectively (orange and purple lines). The blue line indicates the double-logistic model (Eqn. 1) fitted to the observed flux time series (black dots). The slope of the green dash lines indicates the rate of change during the flux development/ recession period. The phenological transition points are marked with the vertical dashed lines, and the bootstrap estimates of 90% confidence intervals of these metrics are indicated with the horizontal red error bars about the 7 key transition points.**

Please clarify the displays of 95% CI from the bootstrapping Monte Carlo runs. In figures with red lines (Figure 2, Figure 3) identified in the labels as representing the magnitude of the 95% CI, a reader sees only a lower line, not - as most might expect. - min and max representing plus/minus 95CI. In text (lines 244, 245) authors give uncertainties in terms of days, and therefore of course always positive. But in Figs 2 & 3, one often sees negative values for 95CI and - as mentioned - only a single value rather than a range. Fix the figures, clarify that the values show by red cross hatches represent absolute values, explain how one represents 95CI with only a single line? Change something to clarify.

Reply: "The error bars refer to phenological metrics, which are time points along the X-axis. The span of the red line indicates the entire CI, from the lowest to the highest value along the X-axis. Uncertainty of fluxes (Y-axis) was not discussed in this paper. The CI are also non-symmetrical about the best estimate, as they were determined using bootstrapping. We have noted this in the legend of Figure 3 to indicate to the reader that this is correct, and not an error of alignment in plotting.

We also moved the CI markers from along the X axis onto the fitted model, and think that this will aid in the interpretation of the figure. The same change was made on Figure 2.

And we added in Lines 194-195: The uncertainties of 7 key transition dates are shown in Fig. 2, as red horizontal error bars to indicate the uncertainty intervals."

[Figure]

**Figure 3: Examples of the seasonal dynamics of different fluxes for 10 sites representative of different biomes. One biome, open shrubland was left off because of space limitations on a single page. The blue line indicates the double-logistic model (Equation 1) fitted to the observed flux time series (black dots). The phenological transition points are marked with the vertical dashed lines, and the bootstrap estimates of 90% confidence intervals of these metrics are indicated with the horizontal red error bar for corresponding transition points. Note that the confidence intervals are not always symmetrical to the best estimate.**

[Figure]

---

## Author Response (AR3)

Dear Editor Carlson,

In this version, we added the **(normalized) rate of peak flux period**, as they characterize summer green-down. So, we updated figures 5, 7, and 9 with two additional new sub-panels, which will not alter the conclusions or the main ideas. Furthermore, these two metrics are also introduced in Table 1 and Figure 2.

[Figure]

**Figure 5: The scatter plots of different phenology metrics from RE-NT daily integrated data and RE-NT daily peak data.**

[Figure]

**Figure 7: The scatter plots of seasonality metrics from RE data using daytime and nighttime partitioning methods.**

[Figure]

**Figure 9: The scatter plots of different phenology metrics from GPP data using daytime and nighttime partitioning methods.**